# Double-layer optimization model for integrated energy system under multiple robustness

Xiaobao Yu [ID]*, Wenjing Zhao

School of Economics and Management, Shanghai University of Electric Power, Shanghai, China

* yuxiaobao1222@163.com

## Abstract

Output instability is one of the important constraints limiting the large-scale application of renewable energy. The development of comprehensive energy systems can effectively improve energy utilization efficiency, but there is still a problem of randomness in renewable energy output. The paper conducts research on the uncertainty of distributed energy output and load, constructs a comprehensive energy system optimization model that takes into account the robustness of bilevel programming, and solves the model using the firefly algorithm. The calculation results show that optimizing uncertainty can significantly reduce the actual operating costs of the system, with a maximum reduction of 14.43%. When the distributed wind power interval is within [0190], a dynamic balance between cost and consumption rate can be achieved.

**Data Availability Statement:** All relevant data are within the manuscript and its Supporting Information files.

## 1 Introduction

The large-scale use of wind power and photovoltaic has become the main means to improve the utilization of integrated energy [1]. As an important manifestation of the energy Internet, the integrated energy system of the park contains different energy subsystems of electricity, heat, cooling and gas, which are strongly coupled with each other. However, due to the randomness of renewable energy output based on wind power and photovoltaic, and the randomness of fluctuation of electricity consumption habits, the consumption of renewable energy and the safe and economic operation of the power grid system are directly limited [2]. Therefore, in order to optimize the power supply structure and promote the coordinated and orderly development of renewable energy sources, how to reduce the uncertainty brought by both sides of the source-load brings a very big challenge and concern to the collaborative operation of the integrated energy system in the park.

Current research has been conducted around regional integrated energy systems. Ghaffari Abolfazl and Askarzadeh Alireza [3] present an efficient and robust optimization approach for size optimization of a hybrid system composed of photovoltaic panel, diesel generator and fuel cell. Zhao Yuehao [4] established an optimal dispatching model for the electricity-gas interconnection system by considering gas supply constraints and introducing data-driven chance constrained programming for the current problems such as high flow volatility of natural gas

**Funding:** The paper is supported by and the Shanghai Municipal Social Science Foundation (No.2020BGL032)

**Competing interests:** The authors have declared that no competing interests exist.

systems. Poursmaeil, B et al [5] proposed the optimal schedule of interconnected MEHs based on the robust optimization in the presence of electric vehicles (EVs). Mao [6] proposed a bi-level collaborative operation strategy for multi-energy systems to achieve load shifting. Asl [7] proposed a bi-level distributed energy dispatching framework for collaborative operation of regional integrated energy systems.

To address the problem of collaborative dispatching of integrated regional energy systems taking into account the uncertainty of new energy output, Lv [8] developed a regional integrated energy system optimal dispatching model considering integrated demand response, which improves the wind power consumption capacity of the system; Yan [9] developed a three-level, two-stage regional integrated energy system robust optimization model to accommodate stochastic disruptions in natural gas and power generation and transmission systems caused by extreme weather; Qi [10] solved the uncertainties associated with variable wind and solar power diffusion and load forecasting errors in a chance constrained programming framework; Gao [11] developed an integrated power/gas/heat system optimal dispatch model; Saberi Hossein et al [12] presents a novel probabilistic model for explicitly quantifying the VESS capacity in charging and discharging modes, which can further be optimized by scheduling building loads; Liu Shuqi [13] proposed a regional integrated energy bi-level optimization model considering energy storage; Yao Zhaosheng et al [14] develops a multi-objective robust optimization framework that accounts for the benefits of multiple parties of smart charging and discharging systems and depicts a bounded uncertainty set based on partial statistical information from real data; Cesena Eduardo Alejandro Martinez and Mancarella Pierluigi [15] proposed a robust operational optimization framework for smart districts with multi-energy devices and integrated energy networks; Zhang Tao [16] established a multi-objective optimal dispatch model considering operation cost, carbon emission, and energy utilization efficiency by using the flexible characteristics of the regional integrated energy system; Yu Xiaobao [17] constructed the critical-peak window determination model and CPP multi-objective optimization model, combing the relevant paths of CPP decision-making; Qiu Zhi [18] established a bi-level model based on particle swarm optimization to cope with renewable energy uncertainty and energy price fluctuations. However, the stochastic rule approach presupposes accurate probability distribution information of the uncertainty factor, and the computation scale is huge and the random variables require complete distribution characteristics. The robust optimization method does not need to obtain the accurate probability distribution of uncertain parameters by constructing an uncertain set of uncertain parameters, and the scheduling optimization results made by this method are too conservative and do not make use of some accessible probabilistic statistical information.

Based on the existing research, an integrated energy system optimization model with bi-level programming robustness is proposed for distributed energy wind power, photovoltaic output and load uncertainty. The upper model takes the minimum operating cost of the electric energy subsystem and the highest renewable energy consumption rate as the multi-objective functions, describes the uncertainties of wind power, photovoltaic and load in the electric energy subsystem in the form of intervals, sets up a game between the system decision maker and the renewable energy output decision maker, and obtains the optimal collaborative solution of the upper model by solving the Nash equilibrium. The lower model is modeled by stochastic chance constrained programming theory, and the uncertainty of energy price is considered to describe the objective function and constraints in probabilistic form.

Therefore, the innovations of this paper include the following three points:

First, based on the bilateral uncertainty of integrated energy system, an optimization model of integrated energy system is constructed, considering the robustness of bi-level programming.

Second, considering the output uncertainty, a multi-objective optimization model of integrated energy subsystem is constructed based on Nash equilibrium.

Third, considering the uncertainty of load and energy price, an integrated energy subsystem optimization model is constructed based on stochastic chance constrained programming.

## 2 Method theory and model construction

### 2.1 System structure

This paper focuses on the uncertainty characteristics of source-side output and user-side load in an integrated energy system. It addresses three key modules: the electric energy subsystem, thermal energy subsystem, and gas energy subsystem [19]. The electric energy subsystem incorporates wind power, photovoltaic renewable energy, and the external power grid, along with demand-side management resources for grid-connected operation. The thermal energy subsystem includes the external heat network, heat storage device, waste heat boiler for heat supply, and the use of dual-effect absorption units to meet user-side cooling load requirements. The gas energy subsystem comprises the external gas network and micro gas turbine for gas supply. The operating structure is illustrated in Fig 1.

Fig 1 highlights the critical role of the electric energy subsystem in achieving collaborative optimization and multi-energy complementarity within the integrated energy system. It serves as the core network and upper-level model for developing optimal operation plans. Incorporating demand-side management resources enhances system efficiency by enabling source-load coordination with the power side. The uncertainty of grid-connected operation from renewable distributed energy output influences system decisions, necessitating the formation of a robust optimization model in conjunction with the decision maker. Furthermore, the thermal energy subsystem and gas energy subsystem operate as lower-level models, making decisions based on the optimized operation plan devised by the electric energy subsystem in a collaborative manner.

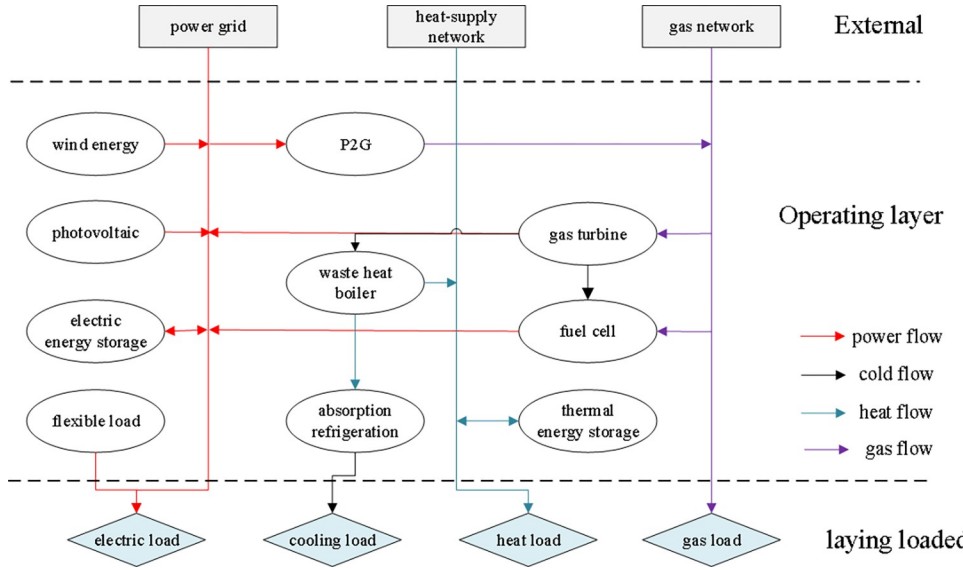

**Fig 1. Schematic diagram of the integrated energy system architecture of the park.**

## 2.2 Model construction

The main parameters and interpretations in the model are shown in Table 1, and will not be repeated in the subsequent model introduction [20].

**2.2.1 Source-load coordination model.** Bi-level optimization theory is a widely adopted approach in decision optimization, effectively addressing the hierarchical nature of decision makers. In a typical bi-level optimization model, control variables are distributed between the upper and lower levels. The upper-level decision maker, acting as the leader, exerts influence over the lower-level decision maker and determines the overall optimization search of the model. The lower-level decision maker's control variables are influenced by the strategy set by the upper-level decision maker. While the objective functions of the upper-level decision maker may be influenced or even determined by the lower-level decision maker, the upper-level decision maker can indirectly impact their own objective functions by influencing the strategies adopted by the lower-level decision maker.

In the source-load collaborative operation structure, the electric energy subsystem plays a crucial role as the core network for energy conversion and transmission. As a result, the source-load coordination plan and optimized operation plan of the electric energy subsystem are of paramount importance within the integrated energy system's optimized operation plan. These plans directly influence the development of the overall source-load operation plan and serve as the upper-level model within the bi-level collaborative optimization model. The operation of other energy subsystems and their interaction with the power subsystem depend on the strategies devised by the upper-level decision makers, thereby serving as the lower-level model within the bi-level collaborative operation model.

**2.2.2 Upper-level optimization model.** *(1) Objective function.* The upper-level model encompasses the source-load collaborative optimization module of the electric energy subsystem. The decision maker's control variables include collaborative plans for distributed generation and load, such as wind power and photovoltaic generation output plans, as well as demand-side management plans. Typically, the upper-level model is formulated as a multi-objective problem. The first sub-objective aims to maximize the consumption rate of wind power and photovoltaic output in the power subsystem. The second sub-objective focuses on minimizing the source-load coordination cost, which comprises the operational cost of the power supply side and the load side's demand-side management cost, namely:

$$\max r = (\sum_{t=1}^{T} P_{\mathrm{WT}}(t) + \sum_{t=1}^{T} P_{\mathrm{PV}}(t)) / (\sum_{t=1}^{T} P_{\mathrm{WT}}^{-}(t) + \sum_{t=1}^{T} P_{\mathrm{PV}}^{-}(t)) \tag{1}$$

$$\min C_{\mathrm{E}} = C_{\mathrm{S}} + C_{\mathrm{L}} + C_{\mathrm{IES}} \tag{2}$$

$$C_{\mathrm{S}} = P_{\mathrm{GRID}}(t)q_{\mathrm{GRID}}(t) + k_{\mathrm{SB}}|P_{\mathrm{SB}}(t)| + k_{\mathrm{WT}}P_{\mathrm{WT}}(t) + k_{\mathrm{PV}}P_{\mathrm{PV}}(t) + f_{\mathrm{MT}}(P_{\mathrm{MT}}(t)) + f_{\mathrm{FC}}(P_{\mathrm{FC}}(t)) \tag{3}$$

$$C_{\mathrm{L}} = c_{\mathrm{delay}}^{0}P_{\mathrm{delay}}^{0}(t) + \sum_{i=1}^{T}[c_{\mathrm{delay}}P_{\mathrm{delay}}(t) + \lambda_{\mathrm{delay}}P_{\mathrm{delay}}(t)] \tag{4}$$

*(2) Constraint conditions.* **a) Power balance constraint of power subsystem.** The upper-level model, serving as the source-load collaborative optimization component of the electric energy subsystem, focuses on developing a source-load coordination plan that ensures power balance within the system. Failure to satisfy this constraint would result in power shortages or

**Table 1. Parameters and interpretation.**

| Parameter | Interpretation | Parameter | Interpretation |
|---|---|---|---|
| $r$ | Rate of clean energy consumption | $P_{WT}$ | Wind consumption power |
| $P_{PV}$ | Photovoltaic consumption power | $\bar{P}_{WT}$ | Forecast output of wind power |
| $\bar{P}_{PV}$ | Forecast output of Photovoltaic | $T$ | Maximum run time |
| $C_E$ | Total cost of system operation | $C_S$ | Operating cost of supply side |
| $C_L$ | Management cost of demand side | $C_{IES}$ | Other system operating costs |
| $P_{GRID}$ | External online power purchase | $q_{GRID}$ | Electricity price level |
| $P_{SB}$ | Energy storage charge and discharge power | $k_{SB}$ | coefficient of ES |
| $k_{WT}$ | Coefficient of wind power operation | $k_{PV}$ | Coefficient of photovoltaic operation |
| $f_{MT}$ | Micro turbine cost | $P_{MT}$ | Micro gas turbine power |
| $f_{FC}$ | Fuel cell cost | $P_{FC}$ | Fuel cell power |
| $c_{delay}^0$ | Cost of fixed translation load | $P_{delay}^0$ | Translatable load power |
| $c_{delay}$ | Coefficient of transfer variable | $P_{delay}$ | Load shifting power |
| $\lambda_{delay}$ | Loss coefficient of transfer electricity benefit | $P_H$ | Thermal power purchased online |
| $p_H$ | External system heating price | $P_X$ | Energy storage charge and discharge heat power |
| $k_X$ | Cost of energy storage unit power | $Q_G$ | Purchasing gas power |
| $C_{NG}$ | Natural gas unit price | $\alpha_H$ | Thermal system confidence level |
| $\bar{C}_H$ | Thermal system optimistic value | $\alpha_G$ | Air energy system confidence level |
| $\bar{C}_G$ | Gas system optimism | $\bar{P}_L$ | Demand management after load power |
| $P_{loss}$ | Network loss power | $P_{E,H}$ | Hydrothermal power |
| $P_{E,G}$ | Heat transfer power | $d(j,t)$ | Load Shift Matrix Elements |
| $R_{MT}$ | Rated climbing rate of micro-turbine | $R_{FC}$ | Rated ramp rate of fuel cell |
| $P_{MT}^{min}$ | Micro gas turbine minimum output | $P_{MT}^{max}$ | Maximum output of micro gas turbine |
| $P_{FC}^{min}$ | Minimum output of fuel cell | $P_{FC}^{max}$ | Maximum output of fuel cell |
| $P_{GRID}^{max}$ | Maximum value of purchased electric power | $P_{E,H,max}$ | maximum power of Electric heat transfer |
| $P_{E,G,max}$ | Maximum power of thermal transfer | $S_{SB}$ | Energy storage remaining power |
| $\eta_{dis}$ | Energy storage and discharge efficiency | $\eta_{ch}$ | Energy storage charging power |
| $S_{SB}^{min}$ | Minimum value of energy storage margin | $S_{SB}^{max}$ | Maximum energy storage margin |
| $P_{SB}^{min}$ | Lower limit of ES power | $P_{SB}^{max}$ | Upper limit of ES power |
| $Q_{SB}$ | Energy storage capacity | $P_E^{max}$ | Maximum supply capacity of the electrical system |
| $\bar{P}_{PV}$ | Photovoltaic actual power | $\bar{\bar{P}}_{PV}$ | Photovoltaic predicted power |
| $\hat{P}_{PV}$ | PV power deviation | $\hat{P}_{PV,min}$ | Photovoltaic deviation minimum |
| $\hat{P}_{PV,max}$ | Maximum of photovoltaic deviation | $\bar{P}_{WT}$ | Actual power of wind power |
| $\bar{\bar{P}}_{WT}$ | Wind power forecast | $\hat{P}_{WT}$ | Wind power deviation |
| $\hat{P}_{WT,min}$ | Minimum of wind power deviation | $\hat{P}_{WT,max}$ | Maximum of wind power deviation |
| $\bar{P}_L$ | Actual power of electric load | $\bar{\bar{P}}_L$ | Electric load forecast power |
| $\hat{P}_L$ | Electric load power deviation | $\hat{P}_{L,min}$ | Minimum value of electrical load deviation |
| $\hat{P}_{L,max}$ | Maximum value of Electric load deviation | $C_{he}$ | Heating coefficient of absorption heating |
| $P_{LH}$ | Thermal load power | $X$ | Thermal energy storage device residual energy |
| $\lambda_x$ | Heat loss coefficient | $Q_{min}$ | Minimum of Charge and discharge heat power |
| $Q_{max}$ | Maximum charge and discharge power | $X_{min}$ | Lower limit of waste heat of heat storage device |
| $X_{max}$ | Upper limit of heat storage device waste heat | $P_H^{min}$ | Minimum power for heat purchase |
| $P_H^{max}$ | Maximum power of heat purchase | $G_s$ | Gas storage tank gas release power |
| $P_{E,G}$ | air-to-electricity power | $P_{L,G}$ | Natural gas load power |
| $Q_s$ | Reserve power of gas storage tank | $G_s$ | Gas tank release power |

*(Continued)*

**Table 1.** (Continued)

| Parameter | Interpretation | Parameter | Interpretation |
|---|---|---|---|
| $Q_s^{min}$ | Lower limit of gas reserve power | $Q_s^{max}$ | Upper limit of gas reserve power |
| $G_s^{min}$ | Lower limit of gas storage and release power | $G_s^{max}$ | Upper limit of gas storage and release power |
| $Q_{in}^{min}$ | Minimum power for purchasing gas | $Q_{in}^{max}$ | Maximum power of purchased gas |
| $u$ | Control variable | $\Pr\{\}$ | Probability of an event |
| $f(u,\xi)$ | Target of the model | $\xi$ | Random variables |
| $\beta_j$ | Confidence | $m$ | Uncertain number of constraints |
| $\bar{f}$ | Optimistic value | $n$ | Number of deterministic constraints |
| $\delta$ | Unbalanced power deviation | $\beta_H\ \beta_G$ | Heat/Gas energy confidence |

surpluses, compromising the reliability of system operations. The power balance constraint is established for each time period in the operation cycle of the integrated energy system and can be expressed as follows:

$$P_{GRID}(t) + P_{MT}(t) + P_{FC}(t) + P_{WT}(t) + P_{PV}(t) + P_{SB}(t) \\ = \bar{P}_L(t) + P_{loss}(t) + P_{E,H}(t) + P_{E,G}(t) \tag{5}$$

$$\bar{P}_L(t) = P_L(t) + P_{delay}(t) - \sum_{j=1, j\neq i}^{T} d(j,t)P_{delay}(j) \tag{6}$$

***b) Renewable energy generation output constraints.*** When the actual consumption plan of wind power output and photovoltaic output is developed in the upper-level model, the actual power consumption cannot exceed the ultra-short-term forecast-ed output with specific constraint expressions, namely:

$$0 \leq P_{WT}(t) \leq \bar{P}_{WT}(t) \tag{7}$$

$$0 \leq P_{PV}(t) \leq \bar{P}_{PV}(t) \tag{8}$$

***c) Controllable micro power output constraint.*** In the upper model, the output of the source-load coordination plan controllable micro power supply needs to satisfy the output limit constraint on the one hand and the ramp rate constraint on the other hand, with specific constraint expressions, namely:

$$\begin{cases} P_{MT}(t)(P_{MT}(t) - P_{MT}^{min}) \geq 0 \\ P_{MT}(t) \leq P_{MT}^{max} \\ P_{FC}(t)(P_{FC}(t) - P_{FC}^{min}) \geq 0 \\ P_{FC}(t) \leq P_{FC}^{max} \end{cases} \tag{9}$$

$$\begin{cases} P_{MT}(t) - P_{MT}(t-1) \leq R_{MT}\Delta t \\ P_{FC}(t) - P_{FC}(t-1) \leq R_{FC}\Delta t \end{cases} \tag{10}$$

In addition, the operation of the common coupling point of the electrical energy subsystem as a connection point for exchanging power with the external network also needs to satisfy the power limit constraint, the specific constraint expression, namely:

$$-P_{\text{GRID}}^{\max} \leq P_{\text{FC}}(t) \leq P_{\text{GRID}}^{\max} \tag{11}$$

*d) Energy conversion constraints.* The source-load coordination center of the electrical energy subsystem, as the upper-level model for the bi-level optimization of the entire integrated energy system, develops the electrical and thermal conversion power and electrical conversion power that directly affect the operation plan of the thermal energy subsystem and the operation plan of the gas energy subsystem, with specific energy conversion constraint expressions, namely:

$$P_{\text{E,H}}(t) \leq P_{\text{E,H,max}} \tag{12}$$

$$P_{\text{E,G}}(t) \leq P_{\text{E,G,max}} \tag{13}$$

*e) Energy storage operation constraints.* Energy storage as a commonly used controllable micro power source of the electric energy subsystem, in solving the optimal operation plan, it is necessary to solve both the charging and discharging power of energy storage in each time period and the remaining power of energy storage in each time period, the specific energy storage operation constraint expression, namely:

$$\begin{cases} S_{\text{SB}}^{\min} \leq S_{\text{SB}}(t) \leq S_{\text{SB}}^{\max}; P_{\text{SB}}^{\min} \leq P_{\text{SB}}(t) \leq P_{\text{SB}}^{\max} \\ S_{\text{SB}}(t+1) = \begin{cases} S_{\text{SB}}(t) - P_{\text{SB}}(t)\Delta t/\eta_{\text{dis}} - \Delta t D_{\text{SB}}Q_{\text{SB}}, P_{\text{SB}}(t) > 0 \\ S_{\text{SB}}(t) - P_{\text{SB}}(t)\Delta t\eta_{\text{ch}} - \Delta t D_{\text{SB}}Q_{\text{SB}}, P_{\text{SB}}(t) < 0 \end{cases} \end{cases} \tag{14}$$

*f) Demand side management constraints.* In the source-load coordination optimization of the electric energy subsystem, which includes load translation constraints and interruptible load constraints, the demand-side management constraints need to be satisfied in the demand-side management plan with the specific expressions, namely:

$$0 \leq \bar{P}_{\text{L}}(t) \leq P_{\text{E}}^{\max} \tag{15}$$

$$0 \leq P_{\text{delay}}(t) \leq P_{\text{delay}}^{0}(t) \tag{16}$$

*g) Constraints of distributed energy decision makers.* In the process of gaming between distributed energy decision makers and system decision makers, the strategy of distributed energy decision makers needs to satisfy the strategy set constraint, which is the interval of distributed energy output. Among them, the randomness of distributed photovoltaic output in robust optimization needs to be described in the form of intervals, with the specific expression, namely:

$$\bar{P}_{\text{PV}}(t) = \bar{\bar{P}}_{\text{PV}}(t) + \hat{P}_{\text{PV}}(t) \tag{17}$$

$$\hat{P}_{\text{PV}}(t) \in [\hat{P}_{\text{PV,min}}(t), \hat{P}_{\text{PV,max}}(t)] \tag{18}$$

Similarly, for the randomness of distributed wind power output is described in interval form with specific expressions, namely:

$$\bar{P}_{\mathrm{WT}}(t) = \bar{\bar{P}}_{\mathrm{WT}}(t) + \hat{P}_{\mathrm{WT}}(t) \tag{19}$$

$$\hat{P}_{\mathrm{WT}}(t) \in [\hat{P}_{\mathrm{WT,min}}(t), \hat{P}_{\mathrm{WT,max}}(t)] \tag{20}$$

Similarly, due to the uncertainty of the load level, the interval form is used to describe the specific expression, namely:

$$\bar{P}_{\mathrm{L}}(t) = \bar{\bar{P}}_{\mathrm{L}}(t) + \hat{P}_{\mathrm{L}}(t) \tag{21}$$

$$\hat{P}_{\mathrm{L}}(t) \in [\hat{P}_{\mathrm{L,min}}(t), \hat{P}_{\mathrm{L,max}}(t)] \tag{22}$$

**2.2.3 Lower-level collaborative operation model.** *(1) Objective function.* The decision strategy of the thermal energy subsystem optimization operation center is the operation plan of the thermal energy sub-network with the lowest integrated operating cost of the thermal energy sub-system as the objective function, namely:

$$\min C_{\mathrm{H}} = \sum_{t=1}^{T} [P_{\mathrm{H}}(t)p_{\mathrm{H}}(t) + k_{\mathrm{X}}|P_{\mathrm{X}}(t)|] \tag{23}$$

The decision strategy of the gas energy subsystem optimization operation center is the operation plan of the gas energy subsystem with the lowest integrated operating cost of the gas energy sub-network as the objective function, namely:

$$\min C_{\mathrm{G}} = \sum_{t=1}^{T} [Q_{\mathrm{G}}(t)C_{\mathrm{NG}}(t)] \tag{24}$$

Based on the lower level stochastic chance constrained programming model, the objective function is expressed in the form of confidence, where the objective function of the decision maker of the thermal energy subsystem, the specific expression, namely:

$$\min C_{\mathrm{H}} = \sum_{t=1}^{T} [P_{\mathrm{H}}(t)p_{\mathrm{H}}(t) + k_{\mathrm{X}}|P_{\mathrm{X}}(t)|] \tag{25}$$

$$\Pr\{C_{\mathrm{H}} \leq \bar{C}_{\mathrm{H}}\} \geq \alpha_{\mathrm{H}} \tag{26}$$

Similarly, the objective function of the decision maker for the gas energy subsystem, the specific expression, namely:

$$\min C_{\mathrm{G}} = \sum_{t=1}^{T} [Q_{\mathrm{G}}(t)C_{\mathrm{NG}}(t)] \tag{27}$$

$$\Pr\{C_{\mathrm{G}} \leq \bar{C}_{\mathrm{G}}\} \geq \alpha_{\mathrm{G}} \tag{28}$$

*(2) Constraint conditions.* In the thermal energy subsystem optimization operation center to develop the operation plan, considering some constraints, including the thermal energy

balance of the thermal energy subsystem, the operation of the energy storage device, the relationship between heat charging and discharging and residual heat of the energy storage device, the purchase of heat power from the external network and other constraints, the specific constraint expressions, namely:

$$
\begin{cases}
P_{MT}(t)C_{he} + Q_x(t) + P_H(t) + P_{E,H}(t) = P_{L,H}(t) \\
X(t) = X(t-1) + P_X(t) - \lambda_x \Delta t \\
Q_{min} \leq P_X(t) \leq Q_{max} \\
X_{min} \leq X(t) \leq X_{max} \\
P_H^{min} \leq P_H(t) \leq P_H^{max}
\end{cases}
\tag{29}
$$

When developing the operation plan in the gas energy subsystem optimization operation center, some constraints need to be considered, mainly including the power balance of the gas energy subsystem, the operation of the gas storage tank, the power of gas purchased from the external network, and other constraints. In addition, in order to calculate the cost of the electric energy subsystem and the thermal energy subsystem easily, the amount of gas in the gas energy subsystem is converted into power with the specific expression, namely:

$$
\begin{cases}
Q_{in}(t) + G_s(t) + P_{E,G}(t) = \dfrac{P_{FC}(t)}{\eta_{FC}(t)Q_{LHV}} + P_{L,G}(t) \\
Q_s(t) = Q_s(t-1) + G_s(t) \\
G_s^{min} \leq G_s(t) \leq G_s^{max} \\
Q_s^{min} \leq Q_s(t) \leq Q_s^{max} \\
Q_{in}^{min} \leq Q_{in}(t) \leq Q_{in}^{max}
\end{cases}
\tag{30}
$$

In the lower-level model, the energy price curves contain uncertainty because the energy prices of the thermal energy external network and natural gas external network use the spot price mechanism. To address the uncertainty brought by the above energy prices, this section adopts stochastic chance constrained programming theory to establish the uncertainty form of the lower-level model, which is a typical stochastic chance constrained programming model. Since the objective function of the decision maker of the lower-level model is to minimize the operating cost of its own decision maker, the form of minimizing the optimistic value is appropriate.

$$
\begin{cases}
\min_u \bar{f} \\
s.t. \\
\quad \Pr\{f(u,\xi) \leq \bar{f}\} \geq \beta \\
\quad \Pr\{g_j(u,\xi) \leq 0\} \geq \beta_j, j = 1, 2, \cdots, p \\
\quad h_i(x) \leq 0, i = 1, 2, \cdots, q
\end{cases}
\tag{31}
$$

Since the lower-level model contains two decision makers, there are some differences in the variables of the above stochastic chance constrained programming model for different lower-level decision makers.

The lower model is modeled in the form of stochastic chance constraint programming implies that the power balance constraint cannot hold absolutely, and the model considers the power balance constraint to be satisfied when the unbalanced power deviation is satisfied

within a certain range and the probability of satisfying the range reaches the confidence level of the set constraint.

The power balance constraint on the probabilistic form of the thermal energy subsystem, namely:

$$\Pr\{-\delta \leq P_{MT}(t)C_{he} + Q_x(t) + P_H(t) + P_{E,H}(t) - P_{L,H}(t) \leq \delta\} \geq \beta_H \tag{32}$$

The power balance constraint for the probabilistic form of the gas energy subsystem, namely:

$$\Pr\left\{-\delta \leq Q_{in}(t) + G_s(t) + P_{E,G}(t) - \frac{P_{FC}(t)}{\eta_{FC}(t)Q_{LHV}} + P_{L,G}(t) \leq \delta\right\} \geq \beta_G \tag{33}$$

# 3 Example simulation

China proposes to achieve carbon neutrality by 2060, a decision that requires a shift in China's energy supply model from traditional fossil energy to large-scale renewable energy. The data of a park in the Yangtze River Delta region of China is selected as the simulation basis. The park has more mature small-scale integrated energy systems, including photovoltaic panels, small wind turbines, heat storage tanks, gas storage tanks, fuel cells, micro gas turbines, energy storage equipment, P2G, etc., which can provide better data support for this paper.

## 3.1 Parameter setting

The distributed photovoltaic and wind power output intervals for a typical operating day of this park's integrated energy system are shown in Fig 2, as well as the electric load level intervals, heating load level and gas load level, as shown in Fig 3.

The park system trades energy and electricity with the external grid, using a time-of-use power price mechanism, which is divided into three periods of peak, flat and valley. The peak period mainly includes 9:00–15:00 and 18:00–21:00, and the peak hour electricity price is 1321

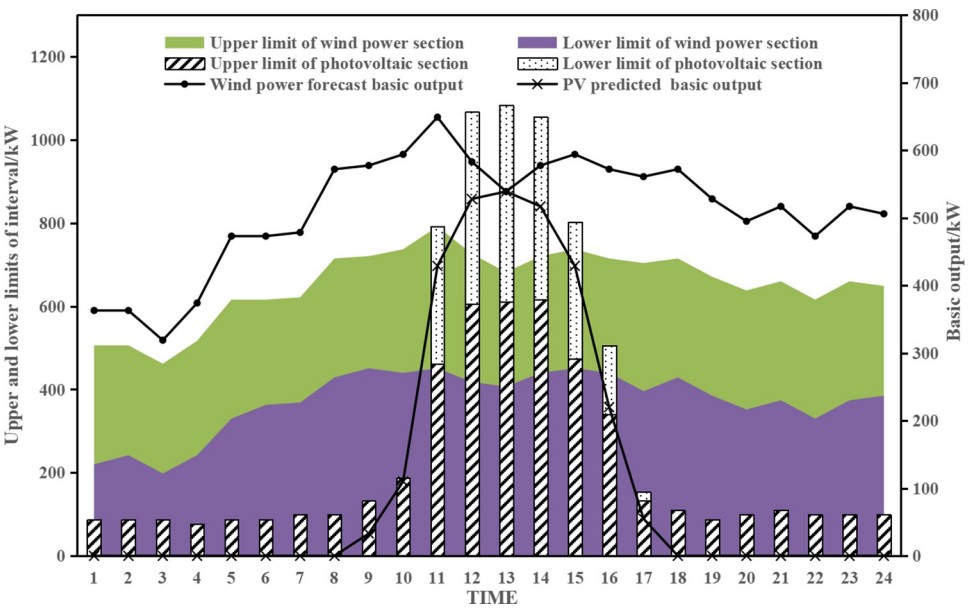

**Fig 2. Clean energy output interval.**

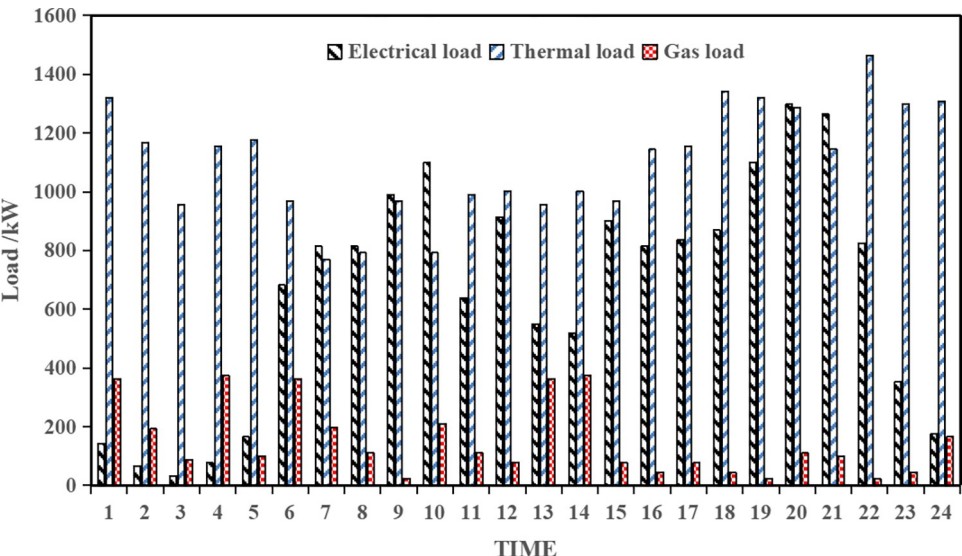

**Fig 3. Electricity load, heat load and gas load requirements.**

yuan/kWh; the valley period mainly refers to 22:00–6: 00, the electricity price is 300 yuan/kWh during off-peak hours; the electricity price is 580 yuan/kWh during other time periods. In addition, the unit price of natural gas in the park is set at 3 yuan/cubic meter, and other parameters are shown in Table 2.

In addition, in the system source-load bi-level optimization model, the model is solved using the firefly algorithm, where the number of fireflies $m$ = 80, the light intensity coefficient $\gamma$ = 1.0, the maximum attraction $\beta_0$ = 1.0, the maximum number of iterations $I$ = 200, the number of chaotic searches is 20 generations, as well as the confidence level in the lower optimization model is taken as 0.95.

## 3.2 Scene setting

As wind power and photovoltaic renewable energy output, and load uncertainty have an important impact on system operation cost and renewable energy consumption. Therefore, how to consider the two-sided uncertainty characteristics for system source-load collaborative optimization operation plan, four different scenarios are set to analyze the park integrated energy system source-load collaborative optimization. Combining with the system source-load collaborative operation structure designed in Fig 1, the base scenario of source-load uncertainty is not considered in scenario A; only the source-side wind power and photovoltaic

**Table 2. Partial parameter settings.**

| number | parameter | dimension | value |
|---|---|---|---|
| 1 | Energy storage charging efficiency | % | 94 |
| 2 | Energy storage and discharge efficiency | % | 95 |
| 3 | Energy storage self-discharge coefficient | / | 0.011 |
| 4 | energy storage capacity | kWh | 1000 |
| 5 | self loss coefficient | / | 0.09 |
| 6 | Maximum capacity of gas storage tank | cubic meter | 400 |
| 7 | Compensation coefficient | / | 2.85 |

output uncertainty scenarios are considered in scenario B: only the load-side various user load uncertainty scenarios are considered in scenario C; and the source-load dual uncertainty scenarios are considered in scenario D.

### 3.3 Operation results

**(1) Collaborative optimization results of scenario A.** According to the source-load collaborative bi-level optimization model, the collaborative optimization results of scenario A can be obtained as shown in Fig 4.

As depicted in Fig 4, the collaborative optimization operation plan developed under scenario A by the integrated energy system effectively meets the energy supply-demand requirements. During time periods 1 to 7, the system purchases power from the external grid due to low electric energy subsystem load and favorable external grid time-of-use prices. Stored power is discharged to cater to peak load hours, reducing reliance on costlier micro-combustion engines and fuel cells. Meanwhile, the thermal energy subsystem procures heat from the external grid for heating needs, with stored heat released through the energy storage device. From the 8th to the 15th

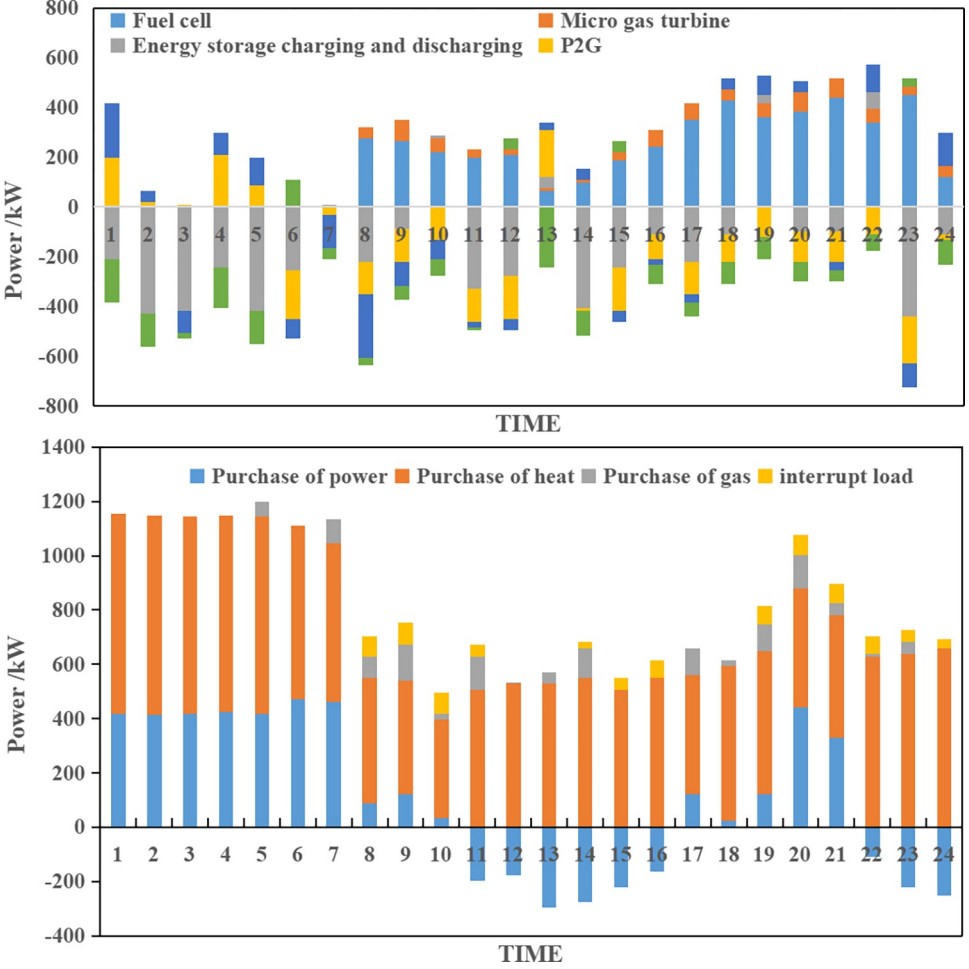

**Fig 4. Integrated energy system collaborative operation optimization plan of scenario A.**

hour, the electric energy subsystem experiences its first load peak, primarily relying on distributed wind power and photovoltaic sources, while also capitalizing on high external grid time-of-use prices to generate revenue from surplus electric energy. During this period, the fuel cell is activated to maximize grid power generation. From the 16th to the 24th hour, the system encounters its second load peak, leading to increased fuel cell and micro-combustion engine usage while fully discharging the energy storage device. Minimizing power purchases from the external grid, the system fulfills power demands through other means, especially as the thermal subsystem load reaches its peak, necessitating additional power procurement. In summary, under scenario A, the integrated energy system is expected to incur a combined operating cost of $2020.06 for a day, achieving an impressive wind and photovoltaic consumption rate of 98.34%.

**(2) Collaborative optimization results of scenario B.** According to the source-load collaborative bi-level optimization model, the collaborative optimization results for scenario B can be obtained as shown in Fig 5.

As depicted in Fig 5, scenario B demonstrates a higher level of controllable micro-power output in the integrated energy system compared to scenario A, which effectively addresses the uncertainty of distributed generation output. The upper-level model considers the uncertainty of distributed wind and photovoltaic output, allowing them to strategic in a way that increases the system's operating cost while reducing the consumption rate of wind power and photovoltaic sources. Additionally, the integrated energy system mitigates the impact of unfavorable renewable energy output by increasing the output of controllable micro-power sources, purchasing more power from the external grid to offset the impact, and strategically incorporating wind and photovoltaic abandonment during certain periods to meet peaking capacity requirements. In summary, under scenario B, the integrated energy system is expected to incur an integrated operating cost of $2,151.38, with a projected integrated consumption rate of wind power and photovoltaic reaching 97.44%.

**(3) Collaborative optimization results of scenario C.** According to the source-load collaborative bi-level optimization model, the collaborative optimization results for scenario C can be obtained as shown in Fig 6.

As depicted in Fig 6, scenario C results in higher expected operating costs for the integrated energy system compared to scenario A due to increased uncertainty in load levels. The system compensates for the adverse effects of uncertainty, albeit sacrificing economy. However, the uncertainty simulation based on the specified system operation plan reveals that the actual average operating costs of the integrated energy system for scenario C are significantly lower than those of scenario A, decreasing by 6.30% from $2273.22 to $2129.98. In summary, under scenario C, the expected integrated operating cost of the integrated energy system is $2228.44, and the actual average expected integrated operating cost of wind power and photovoltaic is $2129.98, with an integrated consumption rate of 96.58%.

**(4) Collaborative optimization results of scenario D.** According to the source-load collaborative bi-level optimization model, the collaborative optimization results of scenario D can be obtained as shown in Fig 7.

As depicted in Fig 7, scenario D has a higher expected operating cost for the integrated energy system, amounting to $2446.54, as it considers both source-side and load-side uncertainties. However, when accounting for the dual uncertainties, the actual average operating cost of the integrated energy system is $2100.69, which outperforms scenarios B and C. In comparison to scenario A, which disregards uncertainties, there is a significant decrease of

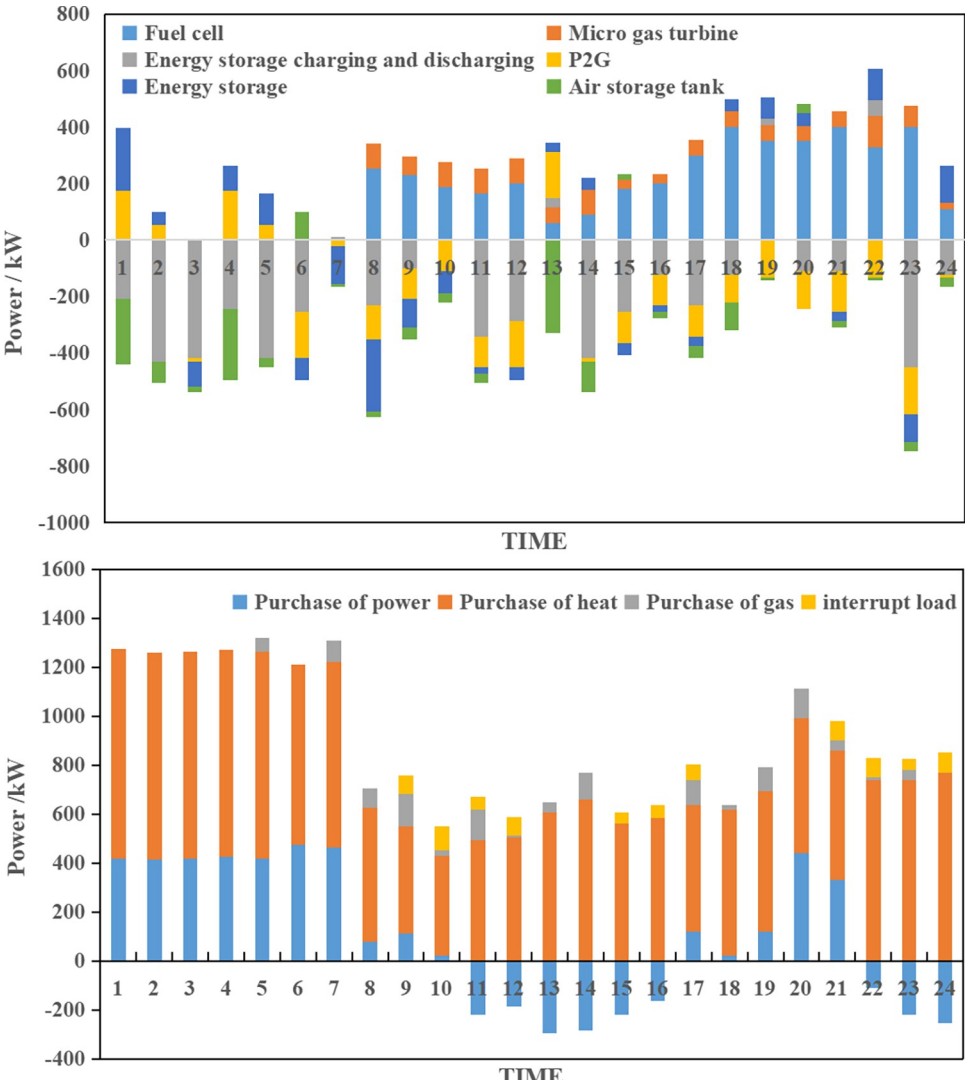

**Fig 5. Integrated energy system collaborative operation optimization plan of scenario B.**

14.43%, validating the correctness and effectiveness of the proposed model. In summary, under scenario D, the expected integrated operating cost of the integrated energy system is $2446.54, the actual average integrated operating cost is $2100.69, and the actual average wind power and photovoltaic consumption rate reaches 97.54%.

## 4 Results analysis

### 4.1 Comparative analysis

Based on the results of the four scenarios mentioned above, the expected integrated operating cost, actual integrated operating cost, and integrated wind photovoltaic consumption rate of the integrated energy system can be obtained. The expected operating cost and expected wind power and photovoltaic consumption rate are derived from the model solution, while the actual integrated operating cost and actual wind power and photovoltaic consumption rate are obtained through 20 simulations considering uncertainties in wind power, photovoltaic, and

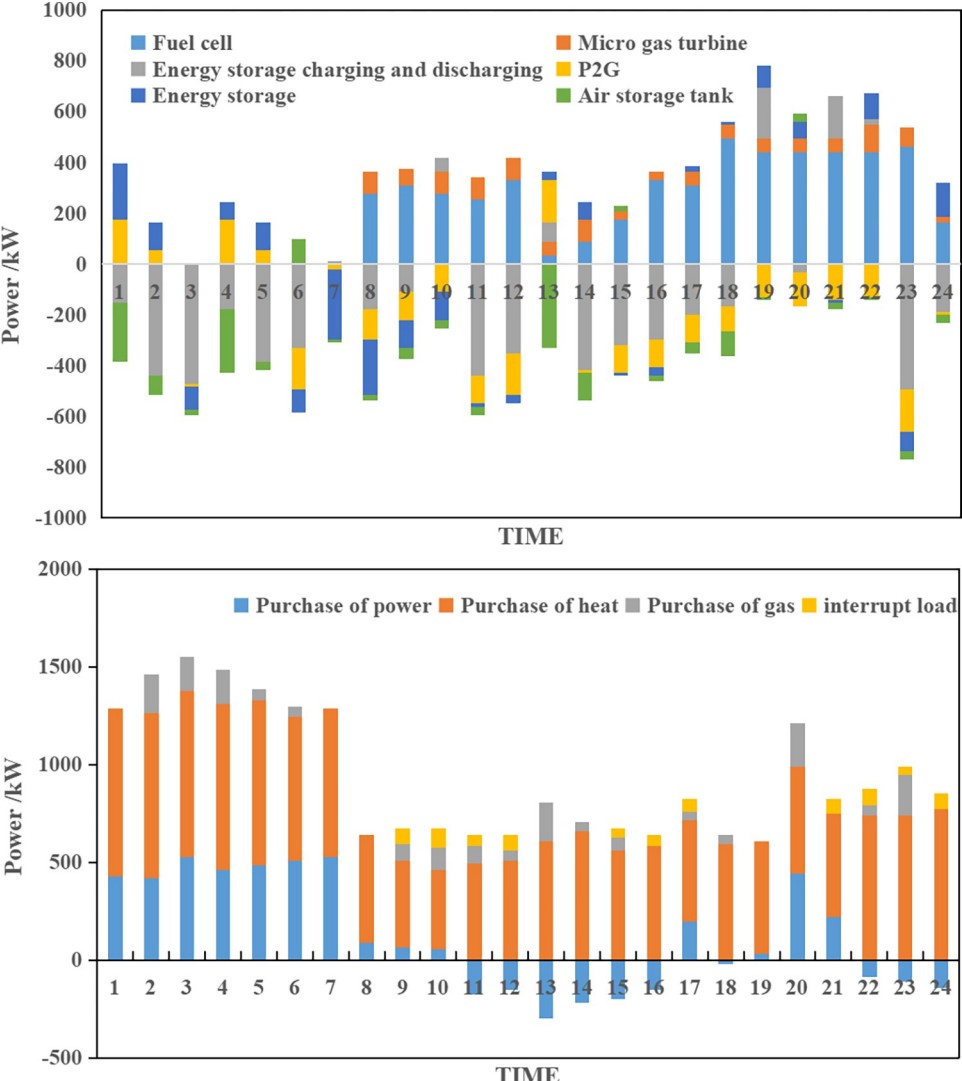

**Fig 6. Integrated energy system collaborative operation optimization plan of scenario C.**

load levels. The average values are calculated based on the developed system operation plan. The specific comparison results are presented in Table 3.

As can be seen from Table 3, under the scenario of system uncertainty, the integrated energy system operation plan developed by the optimization operation center is more applicable to the actual operating conditions, which can significantly reduce the actual operating costs, as well as improve the actual wind power and photoelectric consumption rate.

## 4.2 Sensitivity analysis

In the upper-level robust optimization model, the uncertainty of distributed photovoltaic and wind power output is represented as intervals, which directly impacts the range of strategy sets for distributed wind/photovoltaic and the economic performance of the upper-level model. To illustrate this, Fig 8 shows the relationship between the integrated wind power and photovoltaic consumption rate index and the integrated operating cost for different interval ranges of

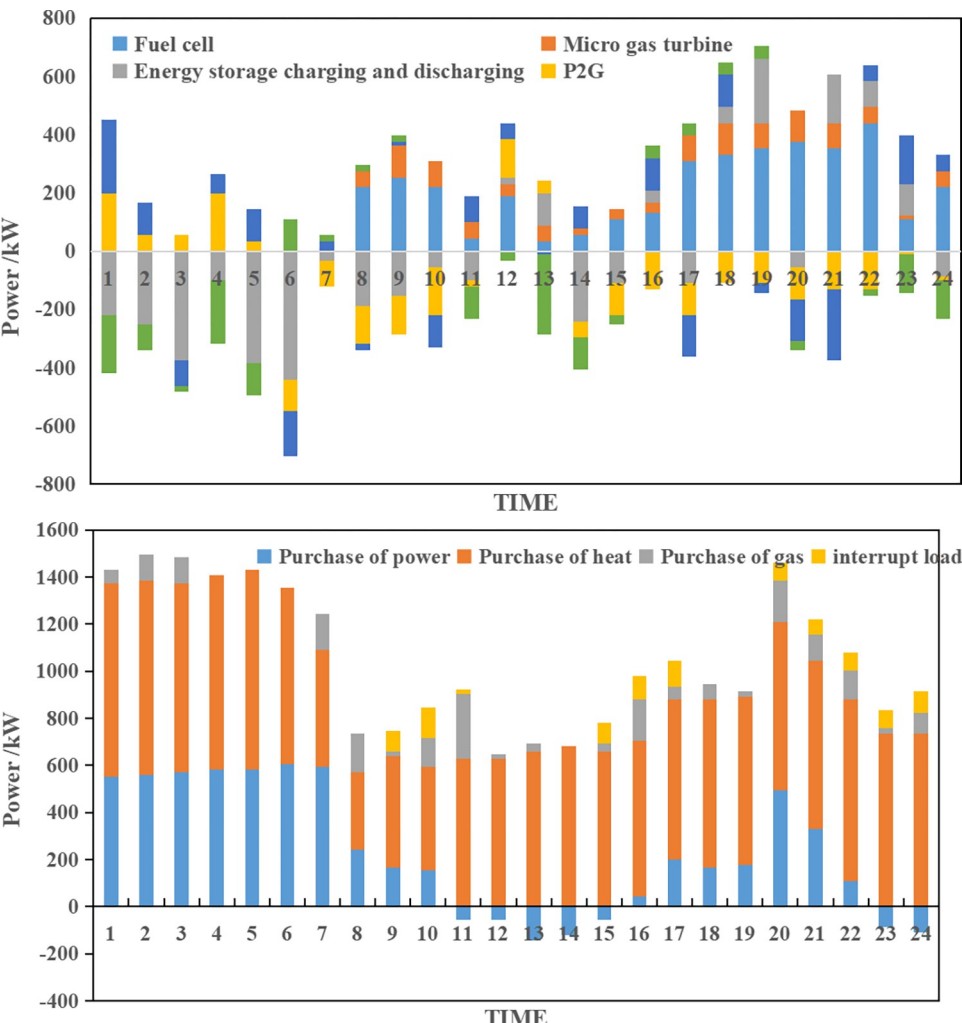

**Fig 7. Integrated energy system collaborative operation optimization plan of scenario D.**

distributed wind power and photovoltaic output. This example focuses on scenario D, which considers double source-load uncertainty.

As shown in Fig 8, the interval range of distributed wind power and photovoltaic output has a significant impact on both the system's wind power and photovoltaic consumption rate index and the upper-level integrated operating cost. As the uncertainty increases, the wind power and photovoltaic consumption rate decreases while the integrated operating cost at the upper level increases. By fitting the curves in the figure, it can be observed that the sensitivity of the wind power and photovoltaic consumption rate to the interval range size is -0.0912/kW, and the sensitivity of the upper-level integrated operating cost to the interval range size is 2.05 $/kW.

**Table 3. Comparison of operating indicators of integrated energy systems.**

| Operation scenario | A | B | C | D |
|---|---|---|---|---|
| Upper model wind power and photoelectric expected consumption rate / % | 98.34% | 97.44% | 96.58% | 93.48% |
| Upper model wind power and photoelectric actual average consumption rate / % | 95.36% | 96.56% | 96.97% | 97.54% |
| Expected operating cost of integrated energy system /$ | 2020.06 | 2151.38 | 2228.44 | 2446.54 |
| Actual average operating cost of integrated energy system /$ | 2273.22 | 2171.39 | 2129.98 | 2100.69 |

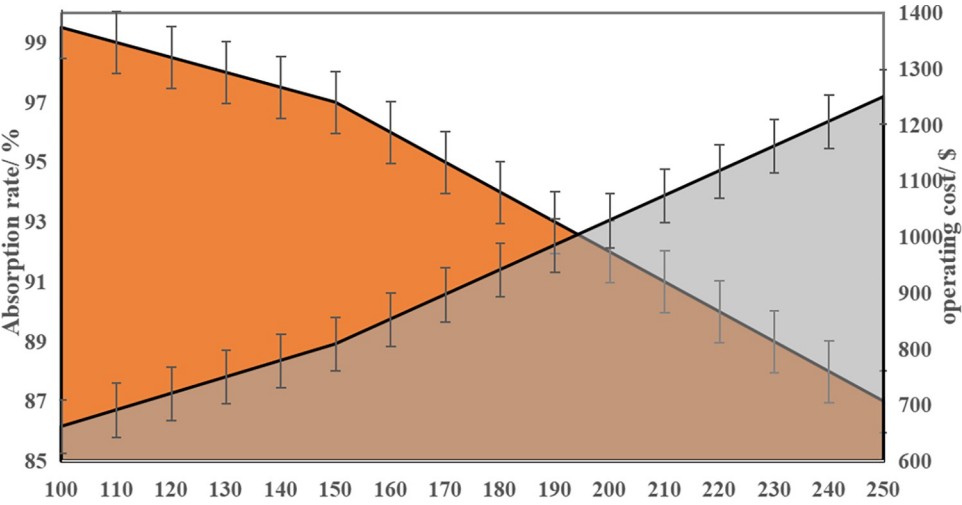

**Fig 8. The sensitivity of the optimal operation index.**

## 5 Conclusion

By carrying out the research on the influence of counting and source-load double uncertainty on the synergistic optimization operation of integrated energy system, the optimization model of integrated energy system is constructed, which can effectively reduce the system operation cost and improve the utilization rate of renewable energy. Based on the establishment of the integrated energy system source-load collaborative bi-level collaborative optimization model and using the firefly algorithm, the following conclusions are obtained:

1. (1) Compared with the scenario in which uncertainty is taken into account, the developed synergistic optimized operation plan of the integrated energy system is more applicable to the actual operating conditions and can significantly reduce the actual operating cost of the system by up to 14.43%;

2. (2) Sensitivity analysis shows that the larger the interval range of distributed wind power, the lower the integrated energy system wind power and photoelectric comprehensive consumption rate, as well as the higher the integrated operating cost, and dynamic equilibrium can be achieved when the interval range is [0,190], when the upper system operating cost is $958.54 and the consumption rate is 93.25%.

## Supporting information

**S1 Data.**
(XLSX)

## Author Contributions

**Conceptualization:** Xiaobao Yu.

**Data curation:** Wenjing Zhao.

**Formal analysis:** Xiaobao Yu.

**Funding acquisition:** Xiaobao Yu.

**Investigation:** Xiaobao Yu.

**Methodology:** Xiaobao Yu.

**Resources:** Wenjing Zhao.

**Writing – original draft:** Xiaobao Yu.

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
