## [Decision Letter · Decision Letter 0]

10 Jul 2023

PONE-D-23-15193Double-layer Optimization Model for Integrated Energy System under Multiple RobustnessPLOS ONE

Dear Dr. Yu,

Thank you for submitting your manuscript to PLOS ONE. After careful consideration, we feel that it has merit but does not fully meet PLOS ONE’s publication criteria as it currently stands. Therefore, we invite you to submit a revised version of the manuscript that addresses the points raised during the review process.

We look forward to receiving your revised manuscript.

Kind regards,

Sani Isah Abba, PhD

Academic Editor

PLOS ONE

“The paper is supported by and the Shanghai Municipal Social Science Foundation (No.2020BGL032)”

“The paper is supported by and the Shanghai Municipal Social Science Foundation (No.2020BGL032).”

“The paper is supported by and the Shanghai Municipal Social Science Foundation (No.2020BGL032)”

Reviewers' comments:

Reviewer's Responses to Questions

**Comments to the Author**

1. Is the manuscript technically sound, and do the data support the conclusions?

Reviewer #1: Partly

2. Has the statistical analysis been performed appropriately and rigorously? 

Reviewer #1: N/A

3. Have the authors made all data underlying the findings in their manuscript fully available?

Reviewer #1: Yes

4. Is the manuscript presented in an intelligible fashion and written in standard English?

Reviewer #1: No

5. Review Comments to the Author

Reviewer #1: The abstract should contain research gaps, novelties, contributions and main results.

For such an international journal, you should provide the research globally not for China. The information about the China should be moved to be as case study.

The research fall into the concepts of "energy management", "Energy efficiency", and "energy awareness". In case these are neglected by the authors, the paper is not on the frontier of the concept and should be purged from publication. a quick scan of the google.scholar with the following keywords, reveals a plenty of research in this regard:

"energy awareness"+"robust optimization" and

"energy efficiency"+"robust optimization"+"smart".

Without considering these papers, the paper should be purged from publication because it is not on the frontier of the concept.

What are the contributions, research questions, and gaps?

You use decentralized system. Why? Using some research papers, you should justify the use of such a system through bi-level programming.

Fter considering such comments, the paper could be further reviewed in case of technical issues.

6. PLOS authors have the option to publish the peer review history of their article (what does this mean?). If published, this will include your full peer review and any attached files.

Reviewer #1: No

<quillbot-extension-portal></quillbot-extension-portal>

---

## [Author Response · Author response to Decision Letter 0]

30 Aug 2023

Response to reviewers:

Reviewer #1:

1. The abstract should contain research gaps, novelties, contributions and main results.

Authors’ Answer: Thanks to the valuable opinions of the experts, the author has rewritten the abstract as follows:

Output instability is one of the important constraints limiting the large-scale application of renewable energy. The development of comprehensive energy systems can effectively improve energy utilization efficiency, but there is still a problem of randomness in renewable energy output. The paper conducts research on the uncertainty of distributed energy output and load, constructs a comprehensive energy system optimization model that takes into account the robustness of bi-level programming, and solves the model using the firefly algorithm. The calculation results show that optimizing uncertainty can significantly reduce the actual operating costs of the system, with a maximum reduction of 14.43%. When the distributed wind power interval is within [0190], a dynamic balance between cost and consumption rate can be achieved.

2. For such an international journal, you should provide the research globally not for China. The information about the China should be moved to be as case study.

Authors’ Answer: Thanks to the valuable opinions of the experts, the author has included the specific situation of China in the case study, please refer to the red section of the paper for details.

3. The research fall into the concepts of "energy management", "Energy efficiency", and "energy awareness". In case these are neglected by the authors, the paper is not on the frontier of the concept and should be purged from publication. a quick scan of the google.scholar with the following keywords, reveals a plenty of research in this regard: "energy awareness"+"robust optimization" and "energy efficiency"+"robust optimization"+"smart". Without considering these papers, the paper should be purged from publication because it is not on the frontier of the concept.

Authors’ Answer: Thanks to the valuable opinions of the experts, the author conducted a literature search based on the combination of "energy awareness" + "robust optimization" and "energy efficiency" + "robust optimization" + "smart", and updated the literature review section. Please refer to the highlighted section of the paper for details.

4. What are the contributions, research questions, and gaps?

Authors’ Answer: Thanks to the valuable opinions of the experts, the contributions of this paper are as follow: based on the bilateral uncertainty of integrated energy system, an optimization model of integrated energy system is constructed, considering the robustness of bi-level programming; considering the output uncertainty, a multi-objective optimization model of integrated energy subsystem is constructed based on Nash equilibrium; considering the uncertainty of load and energy price, an integrated energy subsystem optimization model is constructed based on stochastic chance constrained programming. 

The research questions and gaps are to consider the constraints of uncertainty on the large-scale application of renewable energy, which results in poor performance in comprehensive energy systems. In the context of the contradiction between carbon emissions reduction and system operating costs, how to minimize the uncertainty of supply and demand, and find a dynamic balance between the proportion of renewable energy sources and operating costs is one of the urgent issues to be studied.

5. You use decentralized system. Why? Using some research papers, you should justify the use of such a system through bi-level programming.

Authors’ Answer: Thanks to the valuable opinions of the experts, distributed energy has strong flexibility and is very advantageous for the regulation of integrated energy systems. The paper adopts the double-layer optimization theory to optimize the operation of integrated energy systems considering multiple uncertainties. Currently, this method is a relatively mature theory and can effectively solve multi-objective optimization problems, For example, in the upper level model of the paper, the multi-objective function is to minimize the operating cost of the electrical energy subsystem and maximize the consumption rate of renewable energy. The uncertainty of wind power, photovoltaic power, and load in the electrical energy subsystem is described in interval form, and a game is set up between the system decision-maker and the renewable energy output decision-maker. By solving the Nash equilibrium, the optimal collaborative solution of the upper level model is obtained. The lower level model adopts stochastic chance constrained programming theory for modeling, considering the uncertainty of energy prices, and describing the objective function and constraint conditions in probability form.

---

## [Editor Report · Decision Letter 1]

6 Sep 2023

Double-layer Optimization Model for Integrated Energy System under Multiple Robustness

PONE-D-23-15193R1

Dear Dr. Yu,

We’re pleased to inform you that your manuscript has been judged scientifically suitable for publication and will be formally accepted for publication once it meets all outstanding technical requirements.

Kind regards,

Sani Isah Abba, PhD

Academic Editor

PLOS ONE
---

## [Editor Report · Acceptance letter]

18 Sep 2023

PONE-D-23-15193R1 

Double-layer Optimization Model for Integrated Energy System under Multiple Robustness 

Dear Dr. Yu:

I'm pleased to inform you that your manuscript has been deemed suitable for publication in PLOS ONE. Congratulations! Your manuscript is now with our production department. 

Kind regards, 

on behalf of

Dr. Sani Isah Abba 

Academic Editor

PLOS ONE